# Electrochemical Performance of Potassium Bromate Active Electrolyte for Laser-Induced KBr-Graphene Supercapacitor Electrodes

Nagih M. Shaalan [1,2,*], Faheem Ahmed [1], Shalendra Kumar [1,3], Mohamad M. Ahmad [1,4], Abdullah F. Al-Naim [1] and D. Hamad [2]

1    Department of Physics, College of Science, King Faisal University, P.O. Box 400, Al-Ahsa 31982, Saudi Arabia
2    Physics Department, Faculty of Science, Assiut University, Assiut 71516, Egypt
3    Department of Physics, School of Engineering, University of Petroleum & Energy Studies, Dehradun 248007, India
4    Department of Physics, Faculty of Science, The New Valley University, El-Kharga 72511, Egypt
*    Correspondence: nmohammed@kfu.edu.sa; Tel.: +966-135897114

**Abstract:** In this paper, we have reported a low-concentration active electrolyte of $KBrO_3$ for the supercapacitor's application. The electrochemical processes were carried out in two concentrations of $KBrO_3$ with 0.2 and 0.4 M. Additionally, we have reported a novel strategy for doping graphene during its fabrication process with a potassium bromide (KBr) solution. The chemical doping of graphene with KBr improved the electrochemical properties of graphene used as supercapacitors. HRTEM images confirmed the multi-layer graphene obtained by $CO_2$ laser based on polyimide. The effect of KBr on the graphene lattice has been studied using Raman spectroscopy. The two electrodes of graphene and KBr-doped graphene were subjected to the electrochemical properties study as a supercapacitor by electrochemical impedance spectroscopy, cyclic voltammetry, and galvanostatic charge-discharge techniques. The results exhibited the successful method of graphene doping and the stability of using $KBrO_3$ as a suitable electrolyte for electrochemical processes with this lower molarity. The specific capacitance of the pristine graphene capacitor in 0.2 M of $KBrO_3$ was 33 $Fg^{-1}$, while this value increased up to 70 $Fg^{-1}$ for KBr-doped graphene in 0.4 M of $KBrO_3$. The specific capacity in $mAhg^{-1}$ has also increased twofold. The results exhibited the possibility of using $KBrO_3$ as an electrolyte. The supercapacitor performance almost showed good stability in the life cycle.

**Keywords:** KBr-graphene; laser-induced graphene; supercapacitor; electrochemical; $KBrO_3$ active electrolyte

## 1. Introduction

Supercapacitors (SC) have attracted the attention of researchers because of their high charge and discharge rates, high power ratings, and cycling performance. Capacitors are found in several applications, such as electronic circuits, energy storage, power conditioning, pulsed power, power factor correction, and sensors [1,2]. Supercapacitors are electrochemical energy storage devices with a promising future, which may be able to replace batteries for energy storage in portable electronic circuits and hybrid vehicles. The supercapacitor devices are classified into double-layer electrolytic capacitors (EDLC) [3] and pseudo-capacitors [4]. EDLC capacitors generate their capacitance from the accumulation of charges at the electrode-electrolyte interface. Hence, increasing the surface area and boosting electrical conductivity are useful approaches to realize a high capacity. The energy storage in the pseudo-capacitance is achieved through the transfer of Faraday charges between the electrode and electrolyte with the presence of electrochemical reactions, where part of the energy is not stored in a layer of ions but in chemical bonds [5]. The reversible

multi-electron redox Faradaic interactions are responsible for these interactions, exhibiting a high capacitance and specific energy density.

The SC characteristics are determined by the electrode material and electrolyte type as well. Recently, there have been many extensive studies in this area [6,7]. Graphene has interesting properties, such as being lightweight, having good electrical conductivity, mechanical strength, and having a large surface area [8,9]. Graphene is a 2D monolayer carbon, which is composed of fully sp2 hybridized carbon. Therefore, these properties of graphene have made graphene-based materials find their way into many applications, such as energy devices [10–12]. Despite these outstanding properties, the practical capacitance of pristine graphene is lower than the expected value. Thus, improving the electrochemical performance of graphene remains a major challenge. Graphene has been widely used in materials to support redox reactions. The synergistic effect of the presence of graphene layers in the particles of these materials was observed, and the electrochemical performance was superior to the non-graphene-doped materials.

Several doping methods may affect the electronic properties of graphene; nonetheless, some of them negatively affect its properties, and some of them are complex in their procedure. Therefore, there is still a need for good and easy methods and strategies to improve the electrochemical properties of graphene to make it more effective [13]. Recently, an easy and scalable approach was developed to fabricate 3D porous graphene on polyimide (PI) in one step using a commercial $CO_2$ infrared laser [14]. In this process, laser-induced graphene (LIG) is made in one step, which is an added advantage compared to traditional methods. Few reports have demonstrated direct graphene doping during laser-induced graphene fabrication [15–18].

Reddy et al. [19,20] have reported solid-state electrolytes based on PVC+$KBrO_3$ and PVP+PVA+$KBrO_3$ for solid-state battery applications. The result demonstrated that the magnitude of conductivity increased with an increase in the concentration of the salt. The measurements have shown that the electrolyte is a mixed (ionic + electronic) conductor, the charge transport being mainly ionic. Fic et al. [21] studied the electrochemical performance of a carbon-based supercapacitor in the presence of a bromide/bromate (1.0 M of KBr and 0.05 M of $KBrO_3$) aqueous electrolyte. This study was motivated by the relatively high oxidation potential of the Br, which may be beneficial to the overall cell voltage and capacitance. The study focused on the KBr, $KBrO_3$, and a mixture of KBr and $KBrO_3$. Maher et al. [22] have reported a promising specific capacitance of the activated carbon (AC) electrodes based on potassium bromide (KBr) redox additive electrolyte.

Based on the above sections, substances can affect supercapacitor performance by being present in the electrode or by being a component in the electrolyte. In this paper, we present an active electrolyte of $KBrO_3$ for supercapacitor applications. Electrolyte concentrations of 0.2 and 0.4 M are used for this study. We also present an easy and fast method for making in situ doped graphene. The method is based on a laser fabrication technique and potassium bromide (KBr) solution. The change in the electronic structure of graphene is studied using Raman spectroscopy. The HRTEM investigation is carried out to confirm the multi-layer graphene. A commercial $CO_2$ infrared laser is used for specific power and scan speed. The influence of KBr doping on the electrochemical performance of graphene is investigated. The products are subjected to electrochemical measurements in terms of cyclic voltammetry, galvanostatic charging-discharging, and impedance characteristics.

## 2. Materials and Methods

### 2.1. Materials and Preparations

A potassium bromide (KBr) solution with a molarity of 0.5 M was prepared with deionized water (DI). The graphene sheets were prepared using the $CO_2$ commercial technique. In general, the machine attached has a maximum power of 40 W and 400 mm/s as the maximum speed. The DuPont Kapton Polyimide Films (TapeCase, Elk Grove Village, IL, USA) with a thickness of 170 µm (5.0 mil) were well cleaned. The machine set a power of 8.0 W and a speed of 100 mm/s to form a graphene layer. A 0.5 mL of KBr solution

was dropped on a graphene sheet (10 mm × 50 mm), which dried on a hot plate at 90 °C. The sheet was subjected to the laser beam again under the same conditions. A quantity of a few mg of graphene and KBr-doped graphene powder was collected. Figure 1 shows the procedures for preparing the pristine and KBr-doped graphene composites; based on the experiments, one sample of pristine laser-induced graphene (LIG) and two samples of KBr-doped laser-induced graphene (KBr-LIG). The slurry for electrode synthesis was made by adding 80% of active material, 10% of polyvinylidene difluoride (PVDF) binder, and 10% of carbon black conductive additive to an agate mortar and mixing them with 1-methyl-2-pyrrolidone (NMP) solvent drops. Then, a layer was coated by this slurry on nickel foam with a coating area of 25 $mm^2$ and dried overnight at 80 °C.

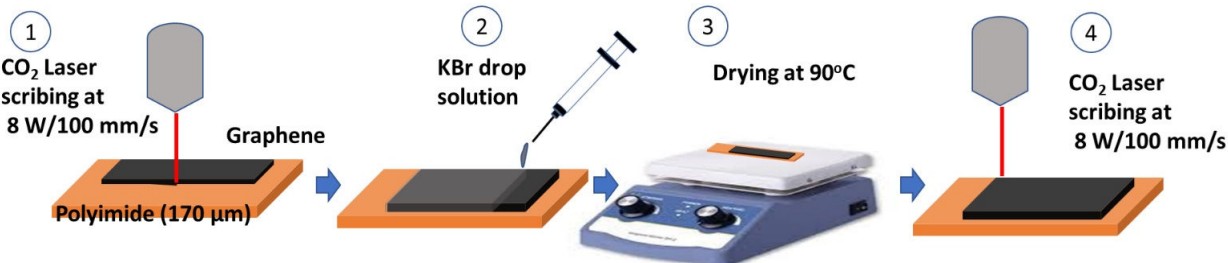

**Figure 1.** The scheme shows the steps for preparing LIG and K-LIG composite by using a $CO_2$ laser machine.

## 2.2. Characterizations

High-resolution transmission electron microscopy (HRTEM: JEOL, JEM-2100F, Tokyo, Japan) was used to analyze the graphene prepared under the current conditions. The sample was dispersed in ethanol and sonicated for 15 min, then 5 μL was deposited on a carbon-coated copper grid, which was dried at low temperature. The equipment works at 200 kV. Raman confocal microscope (LabRAM-HR800) attached to a charge-coupled detector (CCD) was used to investigate the structure phase of the prepared composites using a laser of He-Cd of 633 nm-wavelength and 50 mW-output power at room temperature.

The electrochemical cell contains a counter electrode of Pt wire, a reference electrode of saturated calomel electrode (Ag/AgCl), and an electrolyte of 0.2 and 0.4 M $KBrO_3$ solution. The specific capacitance (*C*) in $Fg^{-1}$ was calculated from the cyclic voltammetry (CV) curve by the formula:

$$C = \frac{A}{2m\,k(V2 - V1)} \tag{1}$$

where *A* is the area of CV in W, *m* is the mass of the active material of 2 mg, *k* is the scan rate in $Vs^{-1}$, and (V2−V1) is the potential window.

The calculated specific capacitance through the galvanostatic charge-discharge (GCD) measurement by the formula:

$$C = \frac{I\,\Delta t}{m\,\Delta V} \tag{2}$$

where *I* is the discharging current density, *Δt* the discharging time, and *ΔV* the drop potential. The electrochemical impedance spectra (EIS) were performed within a frequency range of 1.0 MHz to 0.01 Hz. The SCs measurements were performed using the CorrTest electrochemical workstation System. The CV curves were recorded with scan rates of 5 to 100 $mVs^{-1}$. The GCD curves were carried out from 0.25 to 3.0 $Ag^{-1}$. The stability of electrode performance was tested for 1000 cycles for the charging-discharging curves.

## 3. Results and Discussions

### 3.1. Raman Characterizations

Raman spectra of pristine graphene and KBr-doped graphene samples are shown in Figure 2. The D, G, and 2D bands are observed for Raman spectra, and no other bands

have been observed. The G expresses the graphite bonds present in the graphene and is associated with the $E_{2g}$ vibrations of the $sp^2$ bonds in the Brillouin zone center [23]. Conversely, D expresses the lattice defects of graphene, which can be observed in pristine graphene and appear at 1358 cm$^{-1}$. It is noted from the spectra that the D site changes significantly after the addition of KBr, which indicates the effect of KBr solution in the graphene, causing a modification in the basic configuration of the graphene. The 2D band expresses the second category of the D band. The 2D band is the result of the scattering process of two phonons having opposite wave vectors. Figure S1 shows a more detailed analysis of all the Raman spectra parameters, showing the shift of the three peak positions, the full-width half maximum (FWHM), as well as the area under each peak. Table 1 exhibits the peak position, FWHM, and intensity ratios of the observed bands. The D, G, and 2D bands of pristine graphene are observed at 1358.8, 1568.7, and 2721.6 cm$^{-1}$, respectively. When KBr doping was introduced by laser beam assisting, the D, G, and 2G positions were shifted to 1365.0, 1572.0, and 2733.6 cm$^{-1}$, respectively. The FWHM of the pristine graphene D, G, and 2D bands were 65.3, 46.3, and 117.2, respectively. The peaks became broadened due to KBr-doping, where FWHM became 178.5, 55.6, and 175.7 for D, G, and 2D, respectively. The FWHM increased upon adding KBr, indicating the incorporation of KBr molecules into the graphene lattice. Since the Raman spectra are sensitive to the change of the crystallization process in the material, an increase in the FWHM indicates a disorder in the chains of the graphene lattice. When KBr is dissolved in water, it will dissociate or dissolve into K+(aq) and Br-(aq) ions. These ions bound to the graphene matrix, causing a significant shift in the Raman position of LIG. No other peaks indicate only KBr mode, where KBr has a broadened peak with maximum intensity at 1250 cm$^{-1}$ [24]. Additionally, this can be observed due to the increase in the intensity of the D peak, as well as the increase in the area under peaks, as in Figure 2. The intensity ratio ($I_{D/G}$) of D and G peaks was calculated to be 0.29 for pristine graphene, as it was clearly shown that the prepared pristine graphene contains low lattice defects [25–27]. The ratio of $I_{D/G}$ was to reach 0.38 for graphene doped with KBr. The intensity ratio ($I_{2D/G}$) of the 2D and G peaks was calculated and found to be less than 1.0, confirming the formation of multilayer graphene [27,28]. The value of $I_{2D/G}$ decreased from 0.43 to 0.33 when adding KBr.

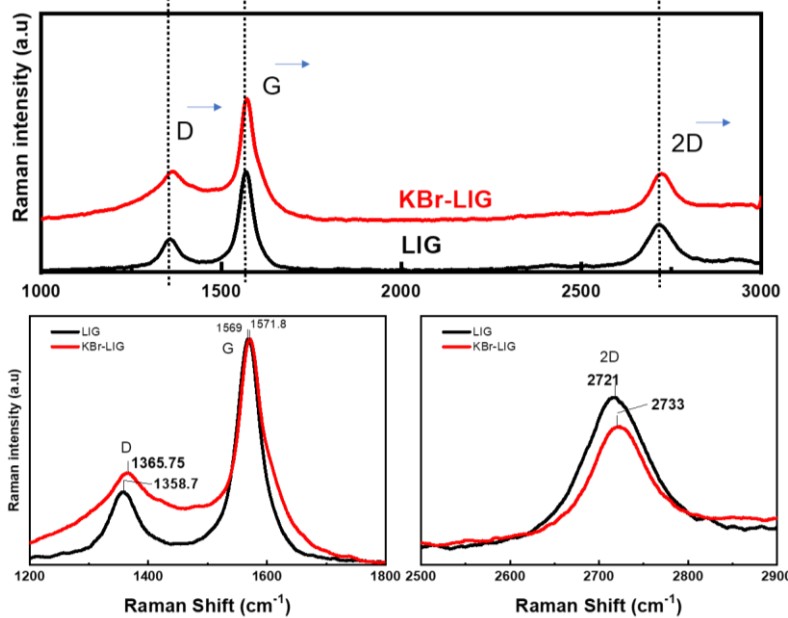

**Figure 2.** The measured Raman spectra of LIG and KBr-LIG samples.

**Table 1.** The peak position, FWHM, and ratios of D to G and 2D to G peaks express the defect- and layer-induced graphene.

| Sample | Peak Position (cm$^{-1}$) | | | FWHM | | | I$_{D/G}$ | I$_{2D/G}$ |
|---|---|---|---|---|---|---|---|---|
| | D | G | 2G | D | G | 2G | | |
| LIG | 1358.8 | 1568.7 | 2721.6 | 65.3 | 46.3 | 117.2 | 0.29 | 0.43 |
| KBr-LIG | 1365.0 | 1572.0 | 2733.6 | 178.5 | 55.6 | 175.7 | 0.38 | 0.33 |

### 3.2. HRTEM Analysis

HRTEM analysis was carried out for the graphene product, as shown in Figure 3. Multilayer graphene nanoribbon-like was observed in the TEM image of Figure 3a. Lattice fringes exhibited a crystalline phase of graphene. The image indicated that the graphene is waved with a width between 3 to 9 nm, as shown in Figure 3b. The inset of Figure 3b is a magnified part of the lattice image where the atomic columns of the carbon are visible. The d-spacing of the graphene lattice planes fabricated here is 0.22 nm. Figure 3c shows the intensity profile of eleven atomic planes, indicating that the d-spacing for these atomic planes is 2.42 nm. The equal vertices of the atomic planes may indicate the straightforwardness of the atomic columns of the carbon alignment.

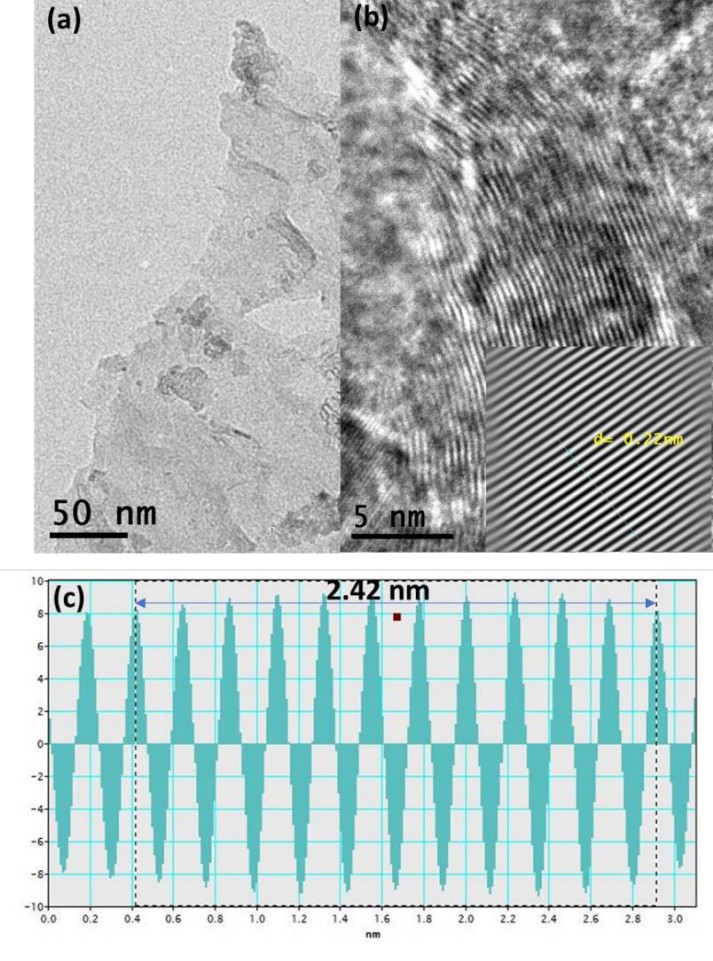

**Figure 3.** High-resolution TEM imaging of graphene. (**a**) TEM image of multilayer graphene, (**b**) lattice image of graphene, the inset is a magnified part where the atomic columns of the carbon are visible, the spacing of these lattice planes is 0.22 nm, and (**c**) The intensity profile shows the spacing between atomic lines, the spacing between 11 atomic planes is ~2.42 nm.

### 3.3. Electrochemical Characterizations

The electrochemical properties of electrodes made of graphene and graphene doped with KBr were measured to evaluate their potential application as supercapacitors in the presence of the novel $KBrO_3$ electrolyte. Figure 4 shows the measured cyclic voltammetry curves of LIG@0.2M and KBr-LIG@0.4M at various scan rates. The potential window is limited between −0.4 to 0.8 V. In the case of LIG@0.2M and KBr-LIG@0.2M, the maximum current density was −3.0 to 3 $Ag^{-1}$ at a scan rate of 100 $mVs^{-1}$, as shown in Figure 4a,b. However, the current density raised two times for the KBr-LIG when measured at 0.4 M of $KBrO_3$, as shown in Figure 4c. We can observe a slight change in the CV shape for KBr-LIG@0.2M, shown in Figure 4b, which was observed for KBr-LIG@0.4M, as shown in Figure 4c. It may exhibit a modification of graphene by KBr, which may modify the electronic structure and electrical properties of graphene [13].

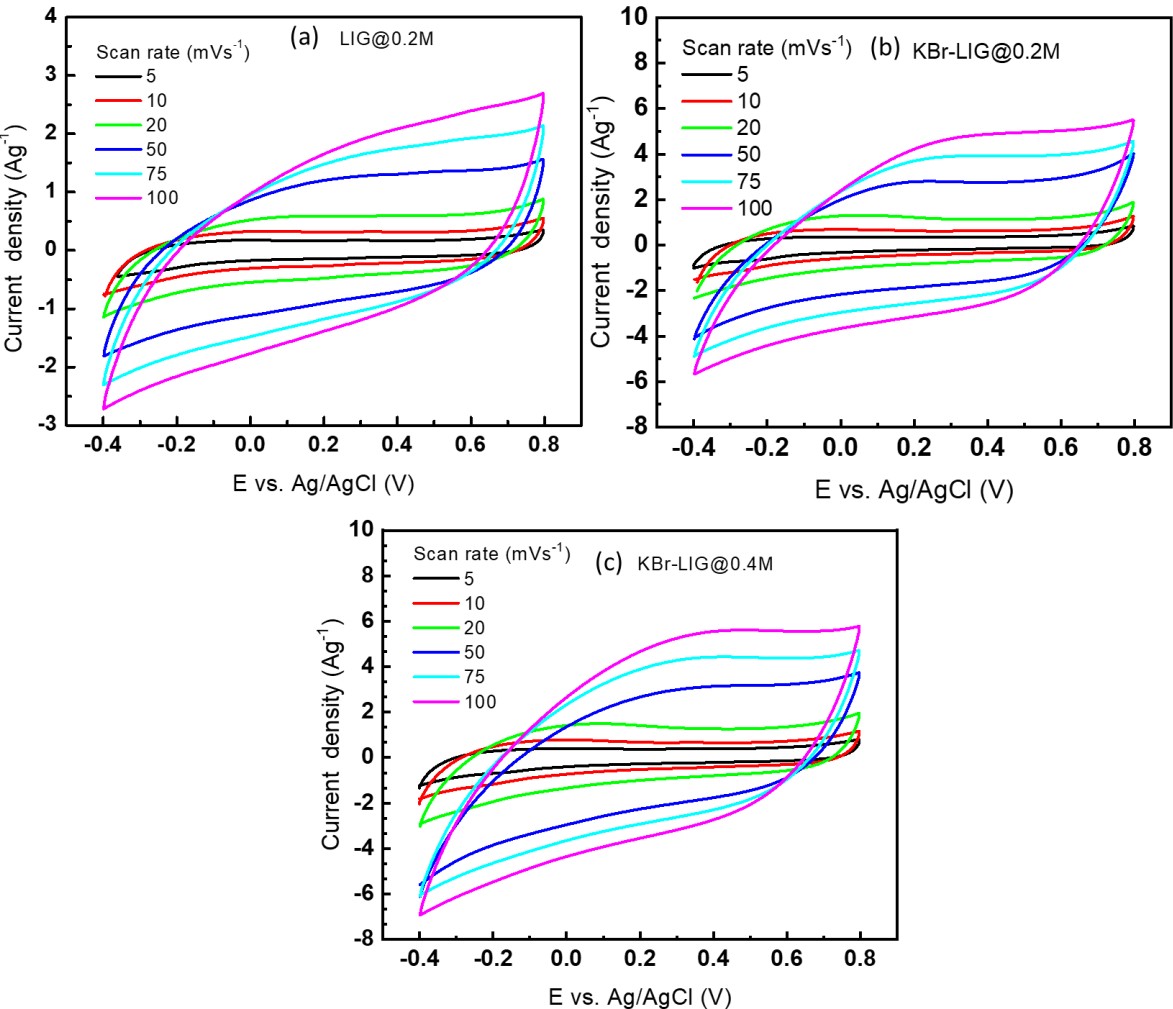

**Figure 4.** Cyclic voltammetry curves measured at different scan rates of (**a**) LIG@0.2M, (**b**), and (**c**) KBr-LIG at 0.2 and 0.4 M of $KBrO_3$ electrolyte.

Figure 5 shows the galvanostatic charging-discharging (GCD) process of the KBr-LIG electrode at two different molarity of the $KBrO_3$ electrolyte. The electrodes were charged and discharged to a voltage of 0.7 V through a current density ranging from 0.25 to 3.0 $Ag^{-1}$. The curves show the possibility of charging the electrodes at a low current density. The full-time of one cycle is 376 s at 0.25 $Ag^{-1}$ for KBr-LIG@0.2M, which was extended to 900 s for KBr-LIG@0.4M, as shown in Figure 5b,c. The full-time cycle at 0.25 $Ag^{-1}$ for LIG@0.2M was 290 s, as shown in Figure 5a. With increasing the current density, the full-time decreased to a lower value. The discharge time of LIG@0.2M was observed between 128 to 6 s when

the current density increased from 0.25 to 3.0 Ag⁻¹. However, the discharging time of KBr-LIG@0.4M was enhanced to 280 to 17 s, indicating an improvement in the specific capacitance. Thus, an increase in the molarity of $KBrO_3$ may enhance the accumulation of charge on the KBr-LIG electrode. However, by increasing the molarity to more than 0.4 M, we found that K and $BrO_3$ ions returned to accumulate to form $KBrO_3$, which was deposited on the cell wall. Thus, we have reached 0.4 M as the maximum soluble molarity.

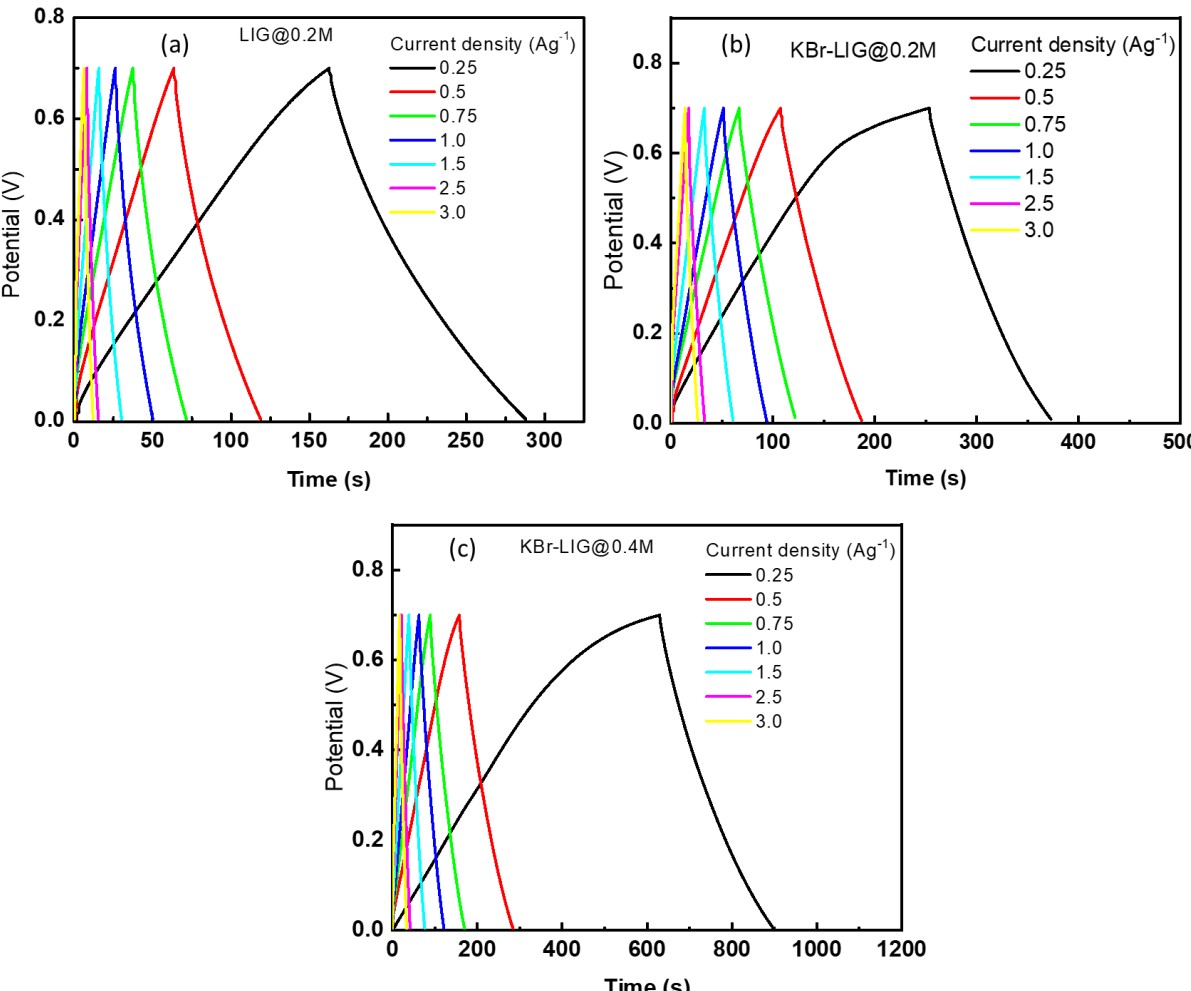

**Figure 5.** Galvanostatic charging-discharging curves as a function of time measured at various current densities for (**a**) LIG@0.2M, (**b**) KBr-LIG@0.2M, and (**c**) KBr-LIG@0.4M.

Based on the previous readings, Figure 6 shows a simple comparison of the electrochemical performance at two different electrolyte concentrations. Figure 6a shows the GCD curves at a low current density of 0.25 Ag⁻¹. The electrochemical performance of KBr-LIG@0.4M is superior compared to LIG@.2M and KBr-LIG@0.2M. Figure 6b shows the CV curves at a low rate of 5 mVs⁻¹. The area included in CV curves is much higher for KBr-LIG@0.4M compared to the others, confirming the higher charge accumulation. Very weak Redox peaks were observed at −0.2 V and 0.3 V for KBr-LIG@0.2M and KBr-LIG@0.2M, respectively. The weak redox peak may be due to modifications of pristine graphene or limited oxidation of graphene. The areas of the CV curves of KBr-LIG@0.2M and KBr-LIG@0.4M to LIG@0.2M are reported in Figure 6b.

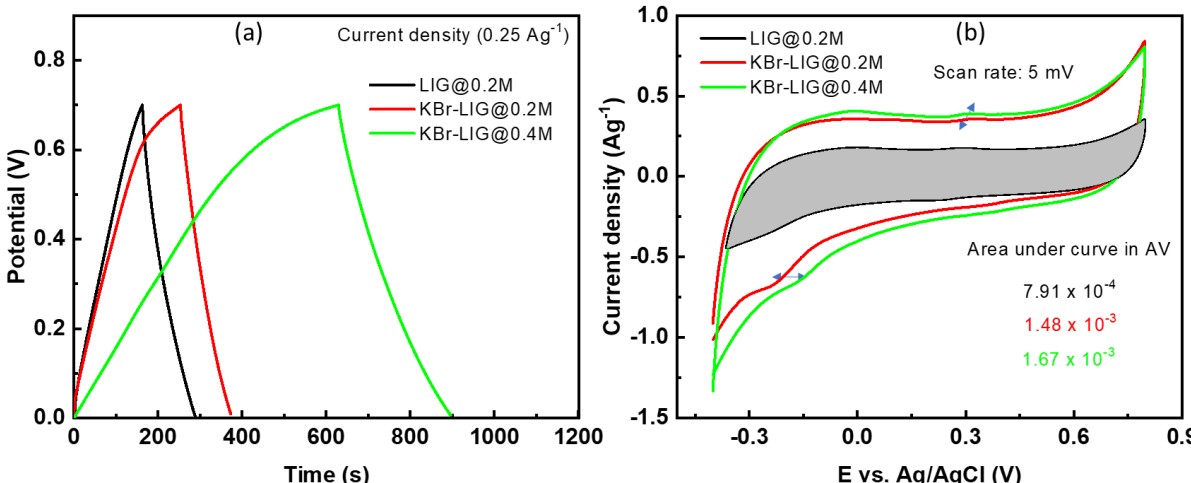

**Figure 6.** (**a**) GCD curves at a current density of 0.25 A$g^{-1}$, (**b**) CV curves at scan rate 5 mV, for LIG@0.2M, KBr-LIG@0.2M, and KBr-LIG@0.4M.

Figure 7 shows the specific capacitance as a function of the voltage and current density of capacitors at two electrolyte concentrations for pristine graphene and doped with KBr. The pristine graphene has a lower specific capacitance than its counterpart doped with the KBr at an electrolyte concentration of 0.2 M. Figure 7a exhibited that the specific capacitance calculated from the CV curves of pristine graphene is limited to between 32 to 12 F$g^{-1}$ at a scan rate confined between 5 to 100 mVs$^{-1}$. Conversely, the specific capacitance of the KBr-doped electrode is 61 to 27 F$g^{-1}$. The capacitance of KBr-LIG was improved based on the 0.4 M concentration of the electrolyte. It was improved up to 70 F$g^{-1}$ at a scan rate of 5 mVs$^{-1}$and 40 F$g^{-1}$ at 100 mVs$^{-1}$. This result was also confirmed in Figure 7b, which shows the specific capacitance versus the discharge current density. The specific capacitance decreases from 23 to 13 F$g^{-1}$ for pristine graphene at a discharge current density of 0.25 to 3.0 A$g^{-1}$. The KBr-LIG also has a specific capacity of 32 to 28 F$g^{-1}$ at an electrolyte concentration of 0.2 M. It is also noted that at 0.4 M, the KBr-LIG has a relatively large specific capacitance between 50 to 40 F$g^{-1}$ at discharge current density of 0.25 to 3.0 A$g^{-1}$.

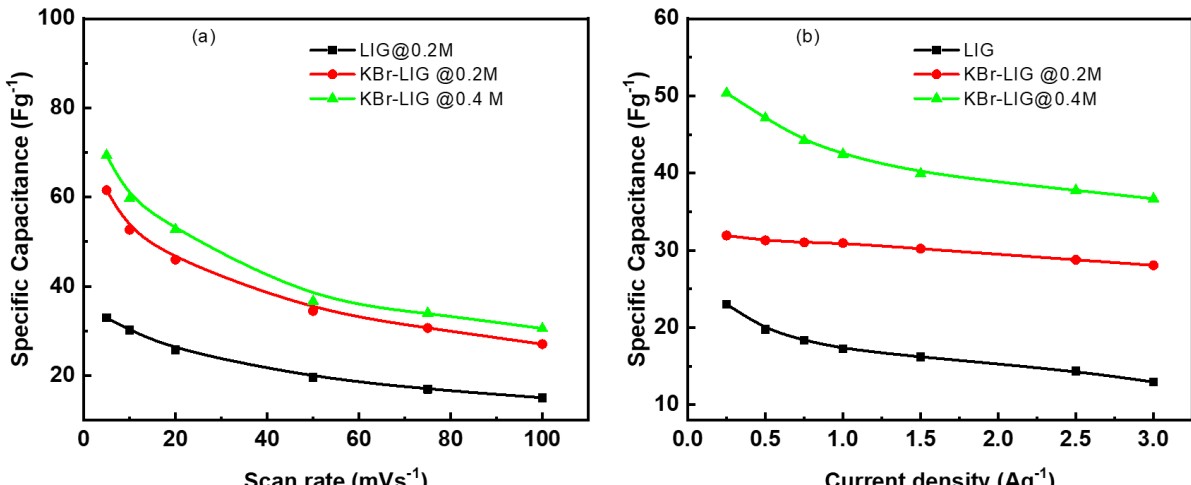

**Figure 7.** (**a**) specific capacitance as a function of scan rate in mVs$^{-1}$, and (**b**) specific capacitance as a function of the current density for pristine LIG, KBr-LIG at 0.2 and 0.4 M of KBrO$_3$ electrolyte.

The corresponding specific energy versus power for LIG and KBr-LIG electrodes are given in Figure 8. The maximum specific energy reported for KBr-LIG is about 3.4 Whkg$^{-1}$ at a current density of 0.25 A$g^{-1}$ and specific power of 43.4 Wkg$^{-1}$, which decreased

to 2.4 Whkg$^{-1}$ at 521.2 Wkg$^{-1}$. The specific power for the electrodes increased from 43.4 Wkg$^{-1}$ to 521.2 Wkg$^{-1}$ when the current increased from 0.25 to 3.0 Ag$^{-1}$. The specific power of LIG and KBr-LIG are the same since the potential and current density is not dependent on the charging or discharging time. Velasco et al. [29] have reported trends in graphene with large areas and micro-supercapacitors. Electrode fabricated by laser direct writing showed various areal or specific capacitances based on the final product obtained either with pristine LIG or doped LIG electrode. In Table 2, a comparison of supercapacitor parameters such as capacitance, energy density, and power was introduced for similar electrodes fabricated by direct laser writing (DLW) on polyimide. The capacitance of LIG was reported between 0.8 mFcm$^{-2}$m up to 995 mFcm$^{-2}$. It was also reported with a value of 115 Fg$^{-1}$ for LIG+PEDOT measured with 1.0 M H$_2$SO$_4$.

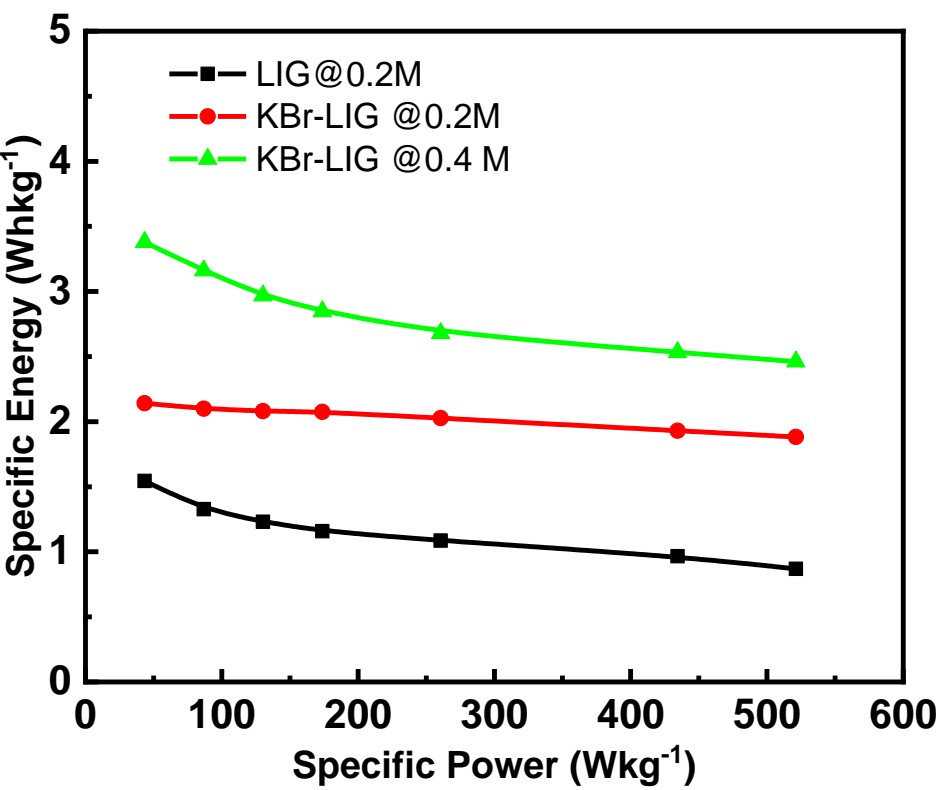

**Figure 8.** Specific energy versus specific power for the supercapacitor for both LIG and KBr-LIG.

**Table 2.** A comparison between different active electrodes prepared by direct laser scribing on a polyimide sheet.

| Active Material | Capacitance | Energy | Power | Electrolyte | Ref. |
|---|---|---|---|---|---|
| LIG | 800 µFcm$^{-2}$ at 10 mVs$^{-1}$ | – | – | PVA–H$_3$PO$_4$ | [30] |
| LIG | 34.7 mFcm$^{-2}$ at 0.1 mAcm$^{-2}$ | 1.0 mWhcm$^{-3}$ | 11 mWcm$^{-3}$ | Li-containing gel electrolyte | [31] |
| LIG | 995 mFcm$^{-2}$ at 1 mAcm$^{-2}$ | 55.9 µWhcm$^{-2}$ | 9.39 mWcm$^{-2}$ | 1 M KOH aqueous | [32] |
| LIG+ MoS$_2$+MnS | 58.3 mFm$^{-2}$ at 50 mAcm$^{-2}$ | 7 µWhcm$^{-2}$ | 49.9 µWcm$^{-2}$ | PVA/Na$_2$SO$_4$ | [33] |
| LIG+PEDOT | 115.2 Fg$^{-1}$ at 0.5 Ag$^{-1}$ | – | – | 1.0 M H$_2$SO$_4$ | [34] |
| NiO/Co$_3$O$_4$/LIG | 29.5 mFcm$^{-2}$ at 0.05 mAcm$^{-2}$ | – | – | PVA–H$_3$PO$_4$ | [35] |
| LIG | 6.1 mFcm$^{-2}$ at 20 mVs$^{-1}$ | 0.96 µWhcm$^{-2}$ | 0.25 mWcm$^{-2}$ | PVA–H$_3$PO$_4$ | [36] |
| LIG | 33 Fg$^{-1}$ at 5 mVs$^{-1}$ | 1.5 WhKg$^{-1}$ at 0.25 Ag$^{-1}$ | 43.3 Wkg$^{-1}$ at 0.25 Ag$^{-1}$to 521 Wkg$^{-1}$ at 3.0 Ag$^{-1}$ | 0.2 M KBrO$_3$ | Present work |
| KBr-LIG | 60 Fg$^{-1}$ at 5 mVs$^{-1}$ | 2.1 WhKg$^{-1}$ at 0.25 Ag$^{-1}$ | | | |
| KBr-LIG | 70 Fg$^{-1}$ at 5 mVs$^{-1}$ | 3.4 WhKg$^{-1}$ at 0.25 Ag$^{-1}$ | | 0.4 M KBrO$_3$ | |

Electrical impedance spectroscopy (EIS) plots for electrodes at electrolyte concentrations of 0.2 and 0.4 M are shown in Figure 9. The equivalent electrical circuit for this set consisting of electrical components was simulated using Zview software. The impedance data were demonstrated by a Nyquist diagram; Z' represents the real impedance, and Z'' is

the imaginary impedance. The Z' signifies the electron transfer resistance (Rct), solution resistance (Rs), and Warburg resistance ($W_{o-R}$). Rs combines the resistance of electrolyte and electrode material, whereas Warburg resistance displays ionic diffusion into the electrolyte, which is frequency dependent. Rct is a function of the electrochemical corrosion reaction intensity at the electrolyte/electrode interface. A higher value of Rct indicates the high integrity of the electrodes and hence a slow progression of corrosion and degradation reactions. The modeled Rs, Rct, and $W_{o-R}$ values are found in Table 3. It is apparent from the Nyquist figure that the semi-circle appears at a higher frequency but with different values. Rs decreased from 10 to 8 Ω when graphene was doped with KBr, indicating the effect of KBr on graphene electronic structure. It also decreased from 8 to 6.5 Ω up to the increase in the molarity of the electrolyte. Rct records values of 18, 18, and 9 Ω for the LIG@0.2M, KBr-LIG@0.2M, and KBr-LIG@0.4M, respectively. $W_{o-R}$ values of 17, 17, and 11 Ω were also recorded for LIG@0.2M, KBr-LIG@0.2M, and KBr-LIG@0.4M, respectively. The decrease in the electron transfer resistance indicates a corrosion process for the KBr-LIG electrode at 0.4 M of electrolyte may occur. The equivalent circuit consisted of $C_{DL}$, which is electric double-layer capacitors (EDLC), where the ions accumulate, forming a double layer of charges at the electrolyte-electrode interface.

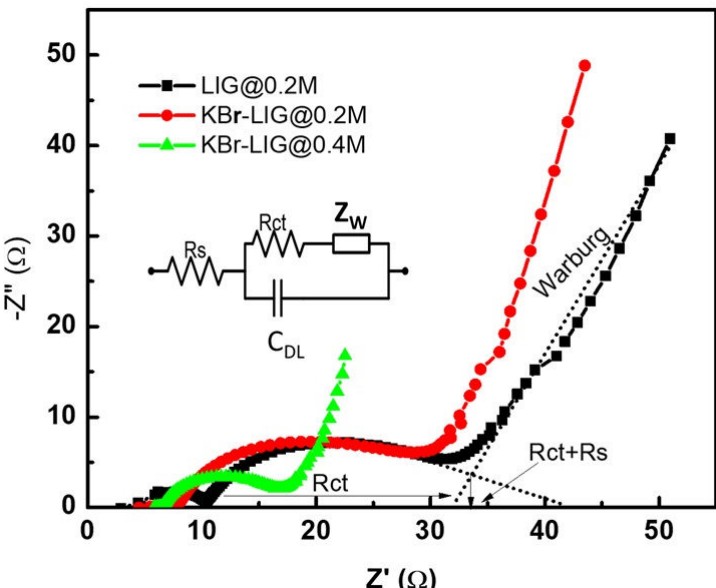

**Figure 9.** Nyquist plot and the equivalent EIS circuit (fitting by Zview software) for LIG@0.2M, KBr-LIG@0.2M, and KBr-LIG@0.4M.

**Table 3.** The value of solution resistance (Rs), electron transfer resistance (Rct), and Warburg resistance for the tested devices.

| Sample | Rs (Ω) | Rct (Ω) | $W_{o-R}$ (Ω) |
|---|---|---|---|
| LIG@0.2M | 10 | 18 | 17 |
| KBr-LIG@0.2M | 8 | 18 | 17 |
| KBr-LIG@0.4M | 6.5 | 9 | 11 |

The stability of the electrochemical process of the electrodes and the electrolyte is an important factor. The retention curves of the specific capacitance value as a function of the number of cycles are shown in Figure 10. The charge and discharge curves corresponding to these values were measured at a current density of 0.75 A$g^{-1}$. It is noted that the electrodes show stability over all the charge and discharge cycles. It was observed that the retention values kept close to 100% with the number of cycles. This means that the specific capacitance of this electrode is also kept whenever this electrode is charged and discharged.

The retention value reached 85% for the KBr-LIG@0.4M, which means a reduction in the specific capacitance of about 15% was observed of its initial value. This behavior may be due to an electrochemical reaction between the electrode material and the KBrO₃ electrolyte. The increase in retention of LIG may be ascribed to the partial oxidation of LIG during the charge-discharge cycles in KBrO₃. The oxidation of the activated carbon surface was performed with a potassium bromate solution [37]. This surface modification of activated carbon with KBrO₃ was carried out in a heated water bath for about 30 min. Lin et al. [38] have reported that carbon dots was oxidized by KBrO₃. Thus, there is a very low level of graphene oxidation expected at elevated voltage during the electrochemical process, where it has been reported that carbon oxidation might occur even below the water decomposition voltage [39], causing an increase in the specific capacitance [40]. It can be observed that this process gradually increases with cycling. A method to determine and find the Faradaic process on carbon electrodes with high surface area by calculating the so-called R-value was proposed [41], which was redefined and later called as S-value (stability value) [42]. The Faraday currents may be masked by the double-layer charge currents that occur at the electrode upon polarization. The literature by Xu [41] proposed that if R-value is above 0.1, more than 10% Faradaic contribution is indicated in the electrolyte decomposition. R-values calculated for LIG@0.2M, KBr-LIG@0.2M, and KBr-LIG@0.4M at a scan rate of 5 mVs⁻¹ were 0.003, 0.138, and 0.154, respectively. These values elucidate the decreased reasons for the electrode retention of KBr-LIG@0.2M and KBr-LIG@0.4M compared to that of LIG@0.2M. However, the KBr-LIG@0.2M electrodes are more stable than KBr-LIG@0.4M, confirming that KBr was bounded within the LIG matrix as K and Br ions, not as KBr molecules, which quickly dissolve in water.

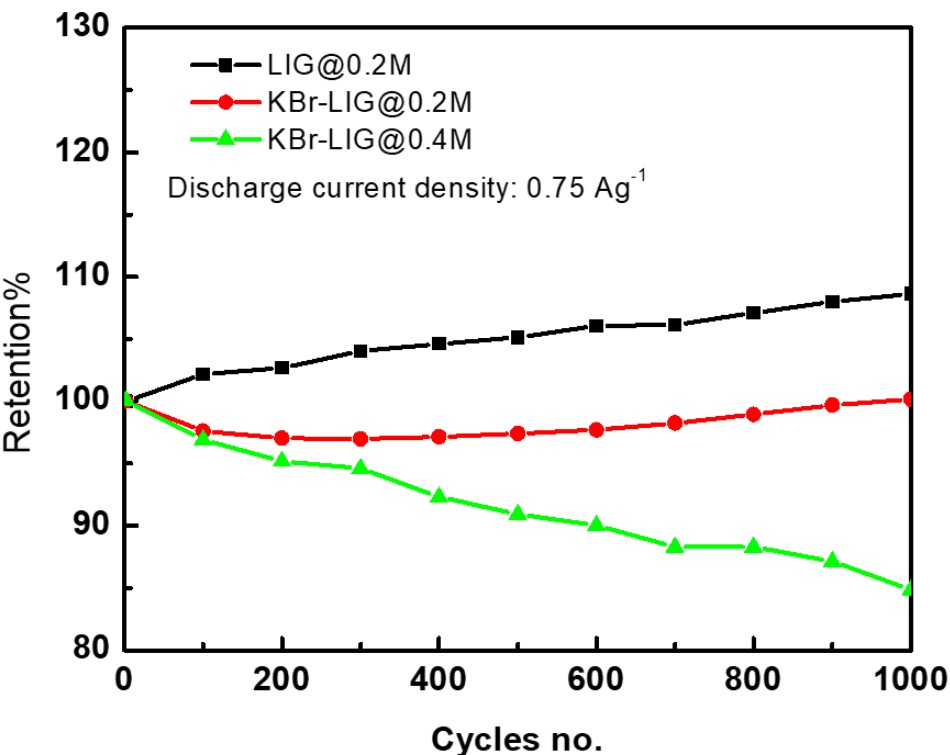

**Figure 10.** Cyclic stability for the specific capacitance retention at a discharge current density of 0.75 Ag⁻¹.

The Coulombic efficiency (CE) is the charge-discharge efficiency calculated by the ratio of discharge to charge capacity. Figure 11a shows the variation of the charge and discharge efficiency versus current density. As observed from Figure 11a, the CE increased with increasing current density. The CE was about 80% and reached 100% for the LIG@0.2M, while it was 45% and 70% for KBr-LIG@0.2M and KBr-LIG@0.4M and reached 100% and 107% when the current density changed from 0.25 up to 3.0 Ag¹. This result suggests the

occurrence of slow and irreversible faradic reactions during the slow charge-discharge process. Thus, at high charge-discharge current density, the slow irreversible faradic reaction cannot follow the fast charge-discharge process and causes an increase of CE and a decrease in the specific capacitance [43]. The CE as a function of cycles is shown in Figure 11b. The efficiency was calculated for the GCD curves obtained at a current density of 0.75 A$g^{-1}$. The CE of the LIG@0.2M started at 91% and enhanced at the first 100 cycles up to 97%. It started with 97% for the KBr-LIG@0.2M and then maintained close to 100% at the first 100 cycles. It recorded a value of 100% and was reduced to 85% for KBr-LIG@0.4M. However, the electrodes still exhibited a good performance toward the electrochemical in terms of the charging and discharging energy. This behavior is due to the precipitation of solid KBrO$_3$ on the electrode surface at 0.4 M concentration of KBrO$_3$, especially at elevated voltage. Fic et al. [21] have determined that the maximum concentration of KBrO$_3$ under the experimental conditions is close to 0.4 mol$L^{-1}$. Thus, the precipitation of salt on the electrode is expected, causing irreversible reactions with the electrode, as indicated by R-value. This also affected the specific capacitance, which decreased with increasing the number of cycles, where the retention of KBr-LIG@0.4M decreased to about 85% at 1000 cycles. Moreover, the Rct at 0.4 M of electrolyte reduced to half of its value, indicating a degradation or corrosion of the KBr-LIG electrode at this electrolyte molarity.

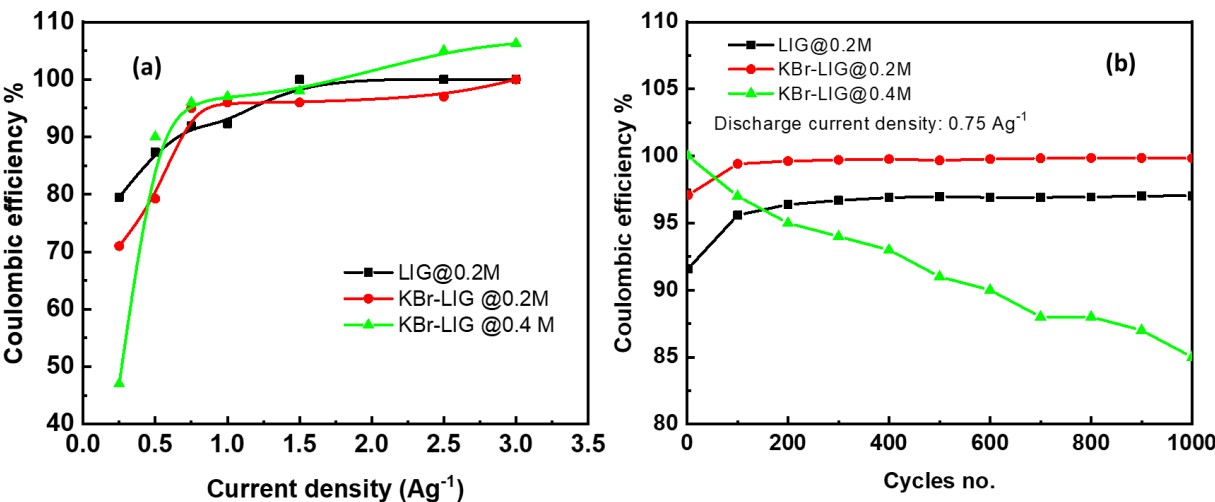

**Figure 11.** The Coulombic efficiency as a function of (**a**) current density and (**b**) cycles for LIG@0.2M, KBr-LIG@0.2M, and KBr-LIG@0.4M.

### 3.4. Electrochemical Mechanism

In the process of electric charging, which occurs by applying a positive electric voltage to the electrode, the electrode tends to collect a positive charge, leading to attracting negative ions present in the electrolyte solution towards it. On the contrary, on the opposite electrode, positively charged ions are attracted due to their negative charge. Therefore, a layer of ions forms on the electrode, resulting in the formation of an electrostatic double layer. Based on the above-reported results, we can perhaps propose a suitable electrochemical mechanism for the present data. KBrO$_3$ is an ionic compound that dissolves in polar solvents like water. According to the Pourbaix diagram of bromine presented by Fic [21], if the pH of the electrolyte is less than 7, bromine ($Br_2$) and polybromides ($Br_3^-$) might br formed. The equations describe these processes are reported for bromide species, as follows:

$$Br^- \leftrightarrow Br_{aq} + e^- \tag{3}$$

$$2Br^- \leftrightarrow Br_{2aq} + 2e^- \tag{4}$$

$$2Br_3^- \leftrightarrow 3Br_{aq} + 2e^- \tag{5}$$

$$Br_{2aq} + 2H_2O \leftrightarrow 2BrO^- + 4H^+ + 2e^- \tag{6}$$

$$Br^- + 2OH^- \leftrightarrow 2BrO^- + 4H_2O + 2e^- \tag{7}$$

$$Br_2 + 2OH^- \leftrightarrow Br^- + BrO^- + H_2O \tag{8}$$

$$BrO^- + 4OH \leftrightarrow BrO_3^- + 2H_2O + 4e^- \tag{9}$$

$$BrO_3^- + 5Br^- + 6H^+ \leftrightarrow 3Br_2 + 3H_2O \tag{10}$$

The pH value of the current solutions ranges between 6.2–6.4. However, due to the absence of bromide and bromate anions in the same electrolyte, the redox chemistry of bromide-based species was hardly induced [21]. According to the electrochemical study by Fic, the bromide species are generated only when KBr and KBrO$_3$ are mixed in the same electrolyte by a molarity ratio of 1:0.05 of KBr: KBrO$_3$. Thus, in the present study, it is expected that bromide species might not exist unless very few amounts of KBr are still attached to graphene in salt form, and it can infiltrate from the electrode to the solution during the electrochemical investigations. This may give a reason for the very weak peak observed at a low scan rate for KBr-LIG electrodes. However, this expected infiltration is an intermediate process that occurred at the beginning. As observed from retention curves, the degradation in capacitance of the KBr-LIG electrode at 0.2 M of KBrO$_3$ can be ignored after 1000 cycles, confirming the stability. In the current study, the performance of electrodes is verified in the presence of $BrO_3^-$ based KBrO$_3$ only. The CV curves of LIG and KBr-LIG suggested that $BrO_3^-$ anion is stable under the conditions and will not impact the electrode performance since there are no redox peaks were observed in the presence of the neutral electrolyte. Thus, the results showed the EDLC behavior of the capacitors. Figure 12 shows the proposed mechanism perhaps explaining this phenomenon based on $BrO_3^-$, and when the electrode is charged, i.e., applying a positive voltage on it, the BrO$_3$ ions move toward it, forming a double layer of charge. In general, supercapacitors use electrostatic phenomena to store energy based on the presence of the electrolyte, which contains positively and negatively charged ions. Some theoretical studies indicated that potassium ions might like to settle in the blank spot of the honeycomb of graphene, which causes an increase in graphene electrons [44,45]. In the CV curve for KBr-LIG@0.2M compared to LIG@0.2M, the electric current density was enhanced slightly. Thus, the present results indicate that chemical doping with KBr may enhance the performance of the graphene-based electrode. Iqbal et al. [13] have confirmed the enhancement in the electronic and electrical properties of a field-effect transistor fabricated by graphene through KBr by the chemical doping method.

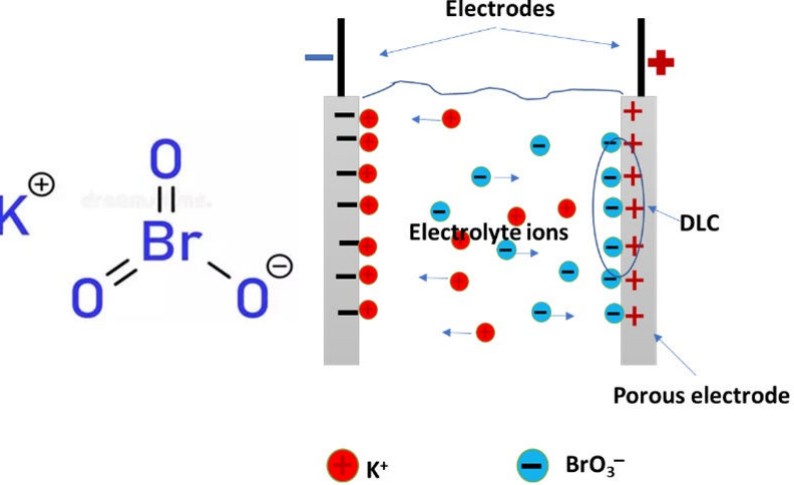

**Figure 12.** Electrochemical reactions mechanism occurs at the electrolyte and electrode interface.



## 4. Conclusions

In summary, a low-concentration active electrolyte was introduced for the supercapacitor application. Additionally, a novel strategy was introduced for doping graphene. The laser-induced graphene was processed with potassium bromide, which was a source of doping. The Raman spectra of the products exhibited the synergistic effect of KBr in the graphene lattice in terms of graphene vibration modes that were modified in position and intensities. The electrolyte with different concentrations was subjected to the electrochemical properties study as a supercapacitor. The electrochemical properties were measured in the presence of $KBrO_3$ electrolyte with 0.2 M and 0.4 M. The incorporation of KBr into the graphene lattice enhanced its electrochemical performance. The results demonstrated that the present method is suitable for doping the graphene, and $KBrO_3$ is suitable as an electrolyte for electrochemical processes. The specific capacitance of the pristine graphene electrode was 33 $Fg^{-1}$ at an electrolyte concentration of 0.2 M, while this value increased up to 70 $Fg^{-1}$ for KBr-doped graphene at an electrolyte concentration of 0.4 M. The cyclic stability confirmed the stable performance of the fabricated supercapacitor over the 1000 cycles investigated here. We could conclude that $KBrO_3$ is a suitable electrolyte for SC and may be extended to be used for other materials and pseudo-capacitors.

**Supplementary Materials:** The following supporting information can be downloaded at: https://www.mdpi.com/article/10.3390/inorganics11030109/s1, Figure S1: The measured Raman spectra of LIG, and KBr-LIG samples.

**Author Contributions:** Conceptualization, N.M.S.; methodology, F.A., N.M.S., S.K. and D.H.; software, F.A.; validation, A.F.A.-N., D.H., S.K. and M.M.A.; formal analysis, M.M.A. and A.F.A.-N.; investigation, N.M.S. and D.H.; resources, N.M.S.; data curation, N.M.S.; writing—original draft preparation, N.M.S. and D.H.; writing—review and editing, M.M.A., S.K. and D.H.; visualization, N.M.S.; supervision, N.M.S.; project administration, N.M.S. and S.K.; funding acquisition, N.M.S. All authors have read and agreed to the published version of the manuscript.

**Funding:** The authors extend their appreciation to the Deputyship for Research & Innovation, Ministry of Education in Saudi Arabia for funding this research work through project number INS136.

**Data Availability Statement:** Available on request.

**Acknowledgments:** The authors extend their appreciation to the Deputyship for Research & Innovation, Ministry of Education in Saudi Arabia for funding this research work.

**Conflicts of Interest:** The authors declare no conflict of interest.

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
