# Peer review of "Electrochemical Performance of Potassium Bromate Active Electrolyte for Laser-Induced KBr-Graphene Supercapacitor Electrodes"

_inorganics, doi:10.3390/inorganics11030109_

Round 1

Reviewer 1 Report

The manuscript reports a study of a supercapacitor made from graphene prepared by laser inducement of polyimide in the presence of KBr. KBrO3 was the electrolyte used.

1     KBrO3 is described as a novel electrolyte on lines 17, 73, 184 and 318 (incorrectly called electrode on 73). It has, however, been used previously together with bromide  in supercapacitor studies such as described in K Fic et al., Redox Activity of Bromides in Carbon-Based Electrochemical Capacitors, Batteries & Supercaps 3 (2020) 1080-1090.

It is pointed out in that article that pH<7 is beneficial to enhance concentrations of species like Br2 and Br3-. No indication of pH is given in the manuscript.

2     Substances can affect supercapacitor performance by being present in the electrode or by being a component in the electrolyte. The manuscript reports KBr incorporated in the graphene electrode and KBrO3 as electrolyte. Sections 3.1 and 3.2 indicate modification of the graphene by the presence of KBr, and there is support for it affecting electrical properties in reference 13, but no evidence is presented in the characterization results how its presence has any significance for supercapacitor performance. Conclusions are drawn about how it affects graphene such as an increase in D peak intensity on line 153 and synergetic effect on 159 but no clarification about its identity in the electrode is given such as if it is still in salt form and strongly bound to the graphene matrix.

It is only electrochemical characterization that gives an indication of the KBr effect on supercapacitor performance and leaching of the salt into the electrolyte may be occurring. Table 2 shows some dopant related differences but there is no discussion as to how the dopant is causing them.

Control experiences are needed where LIG is electrochemically characterized in mixed Br-/BrO3- electrolytes. Low concentrations of KBr added to Na2SO4 electrolyte has been reported, for example, to enhance supercapacitor performance in M Maher et al., Activated carbon electrode with promising specific capacitance based on potassium bromide redox additive electrolyte for supercapacitor application, Journal of Materials Research 11 (2021) 1232-1244.

3     It is not readily obvious that the statement on line 220 that the area of the green CV curve for [email protected] is larger than the red 0.2M one in Figure 6b is correct. The areas look about the same and the percent difference should be given. The percent difference between the black curve shown and a curve for [email protected] should also be reported.

4     Specific energy and specific power for the supercapacitor should be given for both LIG and KBr-LIG. These values along with the given specific capacitances for KBr.LIG should be compared with other graphene materials, such as described in A Velasco et al., Recent trends in graphene supercapacitors: from large area to microsupercapacitors, Sustainable Energy Fuels, 5 (2021) 1235 in order to establish how the work reported in the manuscript is advancing the research field.

5     Equation 3 isn't a very rigorous treatment of the electrochemical mechanism and information in the Fic article in point 1 should be considered as a complement. Especially with regard to clarifying the redox chemistry mentioned on line 191.

Author Response

Dear Respected Reviewer

Thank you very much for your guidance and comments. We are grateful to you for the valuable comments and remarks, which helped improve our work. Below are our point-by-point responses. The corresponding amendments and corrections in the text have been included in the revised manuscript (marked yellow). We hope that our responses satisfactorily address all the issues that you have raised.

Sincerely,

 Nagih Shaalan

Nano and Functional Materials

Reviewer #1

The manuscript reports a study of a supercapacitor made from graphene prepared by laser inducement of polyimide in the presence of KBr. KBrO3 was the electrolyte used.

1     KBrO3 is described as a novel electrolyte on lines 17, 73, 184 and 318 (incorrectly called electrode on 73). It has, however, been used previously together with bromide in supercapacitor studies such as described in K Fic et al., Redox Activity of Bromides in Carbon-Based Electrochemical Capacitors, Batteries & Supercaps 3 (2020) 1080-1090.
It is pointed out in that article that pH<7 is beneficial to enhance concentrations of species like Br2 and Br3-. No indication of pH is given in the manuscript.

Author’s Response: Thank you for your comments.

“electrode” was revised to “electrolyte.”

We agree with the reviewer to point out the useful study reported by Fic et al. Thus, we have pointed out Fic's study in the manuscript.

 Fic et al. studied the electrochemical performance of a carbon-based supercapacitor in the presence of bromide/bromate aqueous electrolytes. In our present study, only KBrO3 with different concentrations was considered. Based on the study, we have proposed an electrochemical mechanism for the current work.   

In the current study, the pH values for the current solutions were 6.2 – 6.4 at a working room temperature of 25oC.

Thank you again for your guidance.

(Please the text, line 79, 381, 402, 415)

2     Substances can affect supercapacitor performance by being present in the electrode or by being a component in the electrolyte. The manuscript reports KBr incorporated in the graphene electrode and KBrO3 as electrolyte. Sections 3.1 and 3.2 indicate modification of the graphene by the presence of KBr, and there is support for it affecting electrical properties in reference 13, but no evidence is presented in the characterization results how its presence has any significance for supercapacitor performance. Conclusions are drawn about how it affects graphene such as an increase in D peak intensity on line 153 and synergetic effect on 159 but no clarification about its identity in the electrode is given such as if it is still in salt form and strongly bound to the graphene matrix.
It is only electrochemical characterization that gives an indication of the KBr effect on supercapacitor performance and leaching of the salt into the electrolyte may be occurring. Table 2 shows some dopant related differences but there is no discussion as to how the dopant is causing them.
Control experiences are needed where LIG is electrochemically characterized in mixed Br-/BrO3- electrolytes. Low concentrations of KBr added to Na2SO4 electrolyte has been reported, for example, to enhance supercapacitor performance in M Maher et al., Activated carbon electrode with promising specific capacitance based on potassium bromide redox additive electrolyte for supercapacitor application, Journal of Materials Research 11 (2021) 1232-1244.

Author’s Response: Thank you very much for this comment. We have added more discussion to the text regarding the point.  We may consider that KBr is bound to the graphene matrix as K and Br ions due to the KBr aqueous and laser scribing. This can be confirmed by Raman spectra as well as the electrochemical performance of the KBr-LIG electrode.

Table 2 became Table 3, and more discussion was added to the Table.

(please see line 83, section 3.1, Table 3, lines 305-3018, line 349)

3     It is not readily obvious that the statement on line 220 that the area of the green CV curve for [email protected] is larger than the red 0.2M one in Figure 6b is correct. The areas look about the same and the percent difference should be given. The percent difference between the black curve shown and a curve for [email protected] should also be reported.

Author’s Response: Thank you very much for this observation. We have recalculated the area under CV curves in AV, as shown in Figure 6. Consequently, the specific capacitance was recalculated. The difference is because the area under CV curves of [email protected] was withdrawn directly from the workstation software, but we have recalculated it by plot software manually, considering the area as shown in Figure 6b for [email protected], as an example.  The area of CV curves is reported in Figure 6b.

(Revised in the text, line Figure 6)

4     Specific energy and specific power for the supercapacitor should be given for both LIG and KBr-LIG. These values along with the given specific capacitances for KBr.LIG should be compared with other graphene materials, such as described in A Velasco et al., Recent trends in graphene supercapacitors: from large area to microsupercapacitors, Sustainable Energy Fuels, 5 (2021) 1235 in order to establish how the work reported in the manuscript is advancing the research field.

Author’s Response:  The corresponding specific energy and power for LIG and KBr-LIG electrodes were given in Figure 8.

Also, a comparison between different active electrodes prepared by direct laser scribing on a polyimide sheet was presented in Table 2.

(Please see the text, Figure 8, line 281)

5     Equation 3 isn't a very rigorous treatment of the electrochemical mechanism and information in the Fic article in point 1 should be considered as a complement. Especially with regard to clarifying the redox chemistry mentioned on line 191.

Author’s Response: To tell the truth we appreciate your guidance through the whole manuscript. Based on your guidance, the manuscript quality was drastically improved.

The electrochemical mechanism was explained based on Fic article information.

(Please see section 3.4, line 401)

We would like to thank you for your guidance, we appreciate that.

Reviewer 2 Report

In the present work, the authors reported an active electrode of KBrO3 for supercapacitor applications. The electrolyte concentrations of 0.2 and 0.4 M are used for this study. The authors presented a method for making in situ doped graphene based on laser-induced graphene and potassium bromide (KBr) solution when the graphene was treated by KBr through a CO2 laser process. The change in the electronic structure of graphene is studied by Raman spectroscopy. The HRTEM investigation is carried out to confirm the multi-layer graphene. A commercial CO2 infrared laser is used for specific power and scan speed. The influence of KBr doping on the electrochemical performance of graphene is investigated. I would recommend this work for publication in Inorganics after a careful revision of the authors. Here are several comments/suggestions:

1.      The number of references reported in the introduction is very less. Authors should add more literature on previous research and identified key gaps.

2.      Carbon based nanomaterials such as graphene in the pristine form are insoluble in DI water and in most organic solvents. In line #110 authors mentioned that “The sample of graphene was sonicated for 5 min, then 5 μL was deposited on a carbon-coated copper grid” Please describe the appropriate method to prepare the graphene sample and the concentration used in the current study.

3.      The full form of HRTEM in line #109 should be High-resolution transmission electron microscopy. Please correct this.

4.      In line #166, authors should not repeat the instrument model names in the discussion section which they already added in the materials and methods section. 

Author Response

Dear Respected Reviewer

Thank you very much for your guidance and comments. We are grateful to you for the valuable comments and remarks, which helped improve our work. Below are our point-by-point responses. The corresponding amendments and corrections in the text have been included in the revised manuscript (marked yellow). We hope that our responses satisfactorily address all the issues that you have raised.

Sincerely,

 Nagih Shaalan

Nano and Functional Materials

Reviewer #2

In the present work, the authors reported an active electrode of KBrO3 for supercapacitor applications. The electrolyte concentrations of 0.2 and 0.4 M are used for this study. The authors presented a method for making in situ doped graphene based on laser-induced graphene and potassium bromide (KBr) solution when the graphene was treated by KBr through a CO2 laser process. The change in the electronic structure of graphene is studied by Raman spectroscopy. The HRTEM investigation is carried out to confirm the multi-layer graphene. A commercial COinfrared laser is used for specific power and scan speed. The influence of KBr doping on the electrochemical performance of graphene is investigated. I would recommend this work for publication in Inorganics after a careful revision of the authors. Here are several comments/suggestions:

Author’s Response: thank you very much for the positive comment.

  1. The number of references reported in the introduction is very less. Authors should add more literature on previous research and identified key gaps.

Author’s Response: Several Refs were added to the introduction and the text body referred to related work. The Refs supported the present study.

  1. Carbon based nanomaterials such as graphene in the pristine form are insoluble in DI water and in most organic solvents. In line #110 authors mentioned that “The sample of graphene was sonicated for 5 min, then 5 μL was deposited on a carbon-coated copper grid” Please describe the appropriate method to prepare the graphene sample and the concentration used in the current study.

Author’s Response:  We agree with the reviewer. This is the method used to upload graphene on a carbon-coated copper grid. The aim is only to upload some of the material on the grid.

 The sample was dispersed in ethanol and sonicated for 15 min, then 5 μL was deposited on a carbon-coated copper grid, which was dried at low temperature.

(Revised in the text)

  1. The full form of HRTEM in line #109 should be High-resolution transmission electron microscopy. Please correct this.

Author's Response: Revised

  1. In line #166, authors should not repeat the instrument model names in the discussion section which they already added in the materials and methods section. 

Author's Response:  Revised

Reviewer 3 Report

This paper report on a new type of laser-induced KBr-graphene for supercapacitor electrodes, along with a novel KBrO3 electrolyte. In comparison to reported laser-induced graphene (LIG) electrode, the author has doped the electrode with KBr, resulting in reduce electrode resistance, therefor the electrode performance got improved.

However, upon reviewing the experiment design and data analysis, I have noticed few issues. The improvement of KBr doping is not significant, with a marginal decrease in electrode resistance (Rs) from 10 ohms to 8 ohms. Furthermore, the benefits and mechanism of KBrO3 electrolyte are not clearly elucidated. As such, I cannot currently recommend the paper for publication in its present form. It may be considered for re-evaluation after the necessary changes been made.

1.       I noticed an inconsistency of the electrolyte concentration used in the main script for the e-chem test of LIG electrode and KBr doped electrode (0.2 M vs. 0.4M) in figure 4 and 5. This inconsistency is potentially misleading for the audience. I would recommend using the same concentration for the electrochemical test to avoid any confusion.

2.       There is significant difference in the potential window used for the CV and GCD electrochemical tests in figures 4 and 5. Is there a specific reason why different potential windows were used for these two tests? It is important to provide a clear explanation to the audience to ensure proper understanding of the experimental setup and results.

3.       Columbic efficiency appears to be low at low current density in GCD curve. Is there an explain for this observation? Additionally, please including a graph depicting the charge/discharge specific capacitance at different current densities or Columbic efficiency plot, as this would enable the audience to compare the rate performance of the electrode more easily.

4.       A minor plot issue on figure 5a 0.75 A/g plot, the curve starting point is not aligned with the rest curve in the plot.

5.       The cycling retention of LIG electrode keep increasing during cycling (around 105% at 1000 cycles), is there an explanation for this phenomenon?

6.       The columbic efficiency of K1.0-LIG electrode in figure 4S keeps decreasing during cycling which suggests that the electrode degradation is accelerating. It would be valuable to explain the mechanism of electrode degradation.

7.       There are few papers reported LIG electrode for supercapacitor. It would be beneficial to provide a comparison of this work with the existing works to evaluate the findings. This comparison could be included in introduction.

Author Response

Dear Respected Reviewer

Thank you very much for your guidance and comments. We are grateful to you for the valuable comments and remarks, which helped improve our work. Below are our point-by-point responses. The corresponding amendments and corrections in the text have been included in the revised manuscript (marked yellow). We hope that our responses satisfactorily address all the issues that you have raised.

Sincerely,

 Nagih Shaalan

Nano and Functional Materials

Reviewer #3

This paper report on a new type of laser-induced KBr-graphene for supercapacitor electrodes, along with a novel KBrO3 electrolyte. In comparison to reported laser-induced graphene (LIG) electrode, the author has doped the electrode with KBr, resulting in reduce electrode resistance, therefor the electrode performance got improved.

However, upon reviewing the experiment design and data analysis, I have noticed few issues. The improvement of KBr doping is not significant, with a marginal decrease in electrode resistance (Rs) from 10 ohms to 8 ohms. Furthermore, the benefits and mechanism of KBrO3 electrolyte are not clearly elucidated. As such, I cannot currently recommend the paper for publication in its present form. It may be considered for re-evaluation after the necessary changes been made.

  1. I noticed an inconsistency of the electrolyte concentration used in the main script for the e-chem test of LIG electrode and KBr doped electrode (0.2 M vs. 0.4M) in figure 4 and 5. This inconsistency is potentially misleading for the audience. I would recommend using the same concentration for the electrochemical test to avoid any confusion.

Authors’ Response: We agree with the reviewer. We have revised the figures to show the CVD and GCD curves for three samples in the same sequences.

(please see Figures 4 and 5)

  1. There is significant difference in the potential window used for the CV and GCD electrochemical tests in figures 4 and 5. Is there a specific reason why different potential windows were used for these two tests? It is important to provide a clear explanation to the audience to ensure proper understanding of the experimental setup and results.

Author’s Response: thank you for this comment.

It is common to find in the literature different values of the working potential window range for aqueous-based supercapacitors. In many cases, even with the best intentions of widening the operating voltage window, the current measured using the cyclic voltammetry (CV) technique includes a significant contribution from the irreversible Faradaic reactions involved in the water splitting process, masked by fast scan rates. Nunes et al. have verified that an apparent potential window of MWCTs was 2.0 V using the CV technique, and it was drastically decreased to 1.2 V after a close inspection of the chronoamperometry findings used to discriminate the presence of a parasitic Faradaic process.

A method to determine and find the Faradaic process on carbon electrodes with high surface area by calculating the so-called R-value was proposed [35], which was redefined and called later as S-value (stability value) [36]. The Faraday currents may be masked by the huge double-layer charge currents that occur at the electrode upon polarization. The literature by Xu [35] proposed that if R-value is above 0.1, more than 10% Faradaic contribution is indicated in the electrolyte decomposition. R-value calculated for [email protected], [email protected], and [email protected] at scan rate of 5 mVs-1 were 0.003, 0.138, and 0.154, respectively. These values suggested the decrease in the electrode retention of [email protected], and [email protected] compared to that of [email protected]

Thus, even the low potential window of GCD of 0.7 V is still high, where the Faradaic process still appears, which can be detected from the low Coulombic efficiency of charge-discharge curves at low current densities, and the retention curves.

(added to the text, line 349)

  1. Columbic efficiency appears to be low at low current density in GCD curve. Is there an explain for this observation? Additionally, please including a graph depicting the charge/discharge specific capacitance at different current densities or Columbic efficiency plot, as this would enable the audience to compare the rate performance of the electrode more easily.

Author’s Response: Thank you for this comment.

(added to the text, line 364)

  1. A minor plot issue on figure 5a 0.75 A/g plot, the curve starting point is not aligned with the rest curve in the plot.

Authors’ Response: Revised.

  1. The cycling retention of LIG electrode keep increasing during cycling (around 105% at 1000 cycles), is there an explanation for this phenomenon?

Author’s Response: Thank you very much for this observation. The increase in retention of LIG may be ascribed to the partial oxidation of LIG during the charge-discharge cycles. The oxidation of the activated carbon surface was performed with a potassium bromate solution [4]. This surface modification of activated carbon with KBrO3 was carried out in a heated water bath for about 30 mins. Lin et al. [5] have reported that carbon dots was oxidized by KBrO3. Thus, there is a very low level of graphene oxidation expected at elevated voltage during the electrochemical process, causing an increase in the specific capacitance [6]. It can be observed that this process gradually increases with cycling since the LIG retention gradually increased.

(Added to the text, line 341)

  1. The columbic efficiency of K-LIG electrode in figure 4S keeps decreasing during cycling which suggests that the electrode degradation is accelerating. It would be valuable to explain the mechanism of electrode degradation.

Author’s Response: we really thank you for this comment, which guides us for more information to improve the manuscript's quality.  

The explanation mechanism of electrode degradation was proposed in the text, line 349 and 379.

  1. There are few papers reported LIG electrode for supercapacitor. It would be beneficial to provide a comparison of this work with the existing works to evaluate the findings. This comparison could be included in introduction.

Author’s Response: A comparison between different active electrodes prepared by direct laser ascribing on a polyimide sheet was added in Table 2.

Thank you again for all your guidance.

Round 2

Reviewer 1 Report

2     The evidence presented is not convincing that K and Br ions in the graphene matrix don't leach out upon immersion in the electrolyte. Raman only shows that their physical presence affects the graphene lattice, not how they're bound to it. A number of control experiments could be done such as re-immersion in a second fresh bromate electrolyte to see if the electrochemistry is the same and Raman measurements after immersion in the electrolyte. But these are something for the future.

3     Percentages on line 253 should be deleted since it is how many times larger the areas are which is shown in Fig 6b.

5     The no KBr in the electrolyte conclusion could be emphasized by showing only equations related to bromate species from the ones on lines 406-413. The KBr concentration in reference 21 was after all 20 times higher than the KBrO3.

Author Response

Reviewer #1

Dear Respected Reviewer

Many thanks to you for your guidance and positive comment. All gratitude to you for the valuable comments and remarks, which helped in improving the paper. The corresponding amendments and corrections in the text have been included in the revised manuscript (marked yellow).

Sincerely,

 Nagih Shaalan

Nano and Functional Materials

Comments and Suggestions for Authors

2     The evidence presented is not convincing that K and Br ions in the graphene matrix don't leach out upon immersion in the electrolyte. Raman only shows that their physical presence affects the graphene lattice, not how they're bound to it. A number of control experiments could be done such as re-immersion in a second fresh bromate electrolyte to see if the electrochemistry is the same and Raman measurements after immersion in the electrolyte. But these are something for the future.

Author’s Response: Thank you very much for this good point. We agree with the reviewer in this regard, we will have this study in the near future.

 3     Percentages on line 253 should be deleted since it is how many times larger the areas are which is shown in Fig 6b.
Author’s Response: Revised

5     The no KBr in the electrolyte conclusion could be emphasized by showing only equations related to bromate species from the ones on lines 406-413. The KBr concentration in reference 21 was after all 20 times higher than the KBrO3.

Author’s response: We agree with the reviewer. We see that it is better to show all equations since they are generally reported for bromine and bromide species to avoid confusion for the reader. We have added the following sentences to clarify the difference in situations.

According to the electrochemical study by Fic, the bromide species are generated only when KBr and KBrO3 were mixed in the same electrolyte by a molarity ratio of 1:0.05 of KBr:KBrO3.

In the current study, the performance of electrodes is verified in the presence of based on KBrO3 only. The CV curves of LIG and KBr-LIG suggested that anion is stable under the conditions and will not impact the electrode performance, since there are no redox peaks were observed in the presence of the neutral electrolyte.

Reviewer 3 Report

Thank you for the revision. I have one comment regarding Figure 8. A dual y-axis plot may not be the most conventional way to present the data in this figure. I would suggest plotting specific energy versus specific power instead.

Author Response

Reviewer #3

Dear Respected Reviewer

Many thanks to you for your guidance and positive comment. All gratitude to you for the valuable comments and remarks, which helped in improving the paper. The corresponding amendments and corrections in the text have been included in the revised manuscript (marked yellow).

Sincerely,

 Nagih Shaalan

Nano and Functional Materials

Comments and Suggestions for Authors

Thank you for the revision. I have one comment regarding Figure 8. A dual y-axis plot may not be the most conventional way to present the data in this figure. I would suggest plotting specific energy versus specific power instead.

Author’s Response: Thank you for your positive comment. Figure 8 was plotted according to the reviewer's guidance.